# Highlighting Thermo-Elastic Effects in Confined Fluids

**DOI:** 10.3390/polym13142378

**Published:** 2021-07-20

**Authors:** Eni Kume, Patrick Baroni, Laurence Noirez

**Affiliations:** Laboratoire Léon Brillouin (CEA-CNRS), Université Paris-Saclay, CEA-Saclay, CEDEX, 91191 Gif-sur-Yvette, France; eni.kume@cea.fr (E.K.); Patrick.baroni@cea.fr (P.B.)

**Keywords:** liquid state, polymer melt, glass former, shear strain, shear elasticity, adiabatic transformation, thermo-elasticity

## Abstract

The recent identification of a finite shear elasticity in mesoscopic fluids has motivated the search of other solid-like properties of liquids. We present an innovative thermal approach of liquids. We identify a dynamic thermo-elastic mesoscopic behavior by building the thermal image produced by different liquids upon applying a low frequency mechanical shear field. We selected three fluids: a low molecular weight polybutylacrylate (PBuA), polypropyleneglycol (PPG), and glycerol. We demonstrate that a part of the energy of the shear strain is converted in cold and hot shear bands varying synchronously with the applied shear field. This thermodynamic change suggests a coupling to shear elastic modes in agreement with the low frequency shear elasticity theoretically foreseen and experimentally demonstrated.

## 1. Introduction

The recent consideration of the fluid/substrate interfacial forces in the protocol of dynamic mechanical analysis has enabled us to demonstrate that the viscoelastic response is not universal but can be modulated by the fluid/surface boundary conditions and the scale at which the fluid response is measured. Indeed, the surface molecules explore other neighborhoods and are in another thermodynamic state with respect to the bulk molecules. This particular interfacial configuration questions the solid-like or liquid-like nature of the fluid, challenges the assumption of uniform viscoelasticity and characteristic relaxation times [1], generates dynamic stick-slip friction, formation of lubrication layer [2], and in extreme situations, autophobicity effects [3], or in contrast, produces a strengthening of the liquid molecule interaction with the solid substrate under strong wetting conditions. It has been shown that the strong liquid/substrate interaction amplifies the fluid response to a mechanical field and has revealed the existence of a mesoscopic “static” shear elasticity (0.1–10 Hz) in various fluids including polymer melts and molecular liquids [4,5,6,7,8,9,10]. The shear elasticity of fluids is a scale dependent property, the force of which increases as the probed scale decreases [11,12,13,14]. Figure 1 illustrates the spectacular transition from viscous (the viscous modulus *G*” scaling with the square of the frequency *ω*^2^) to an elastic-like response (the shear modulus *G*’ dominates and is independent of the frequency) of the mesoscopic mechanical response of the glycerol in standard conditions and when using a high wetting substrate [9].

The experimental identification of a “static” liquid shear elasticity at low scale has received an important echo. This is in agreement with recent theoretical predictions foreseeing that liquids can support the propagation of shear waves above nanoscopic scales, validating the measurement of a solid-like response extended up to a finite length scale of *1/k* where *k* is the wavevector over which shear elastic waves can propagate (*k*-gap model [15,16,17,18,19,20,21]). The liquid shear elasticity has independently established by both an experimental and theoretical approaches; that is, the mesoscale liquid state is dominated by long-range elastic intermolecular interactions [22,23,24,25].

Because of the “static” shear elasticity, fluids resist flow below an elastic threshold whose resistance depends on the considered scale. The mesoscopic shear elasticity concerns both simple liquids (van der Waals and H-bond liquids), complex fluids (polymer melts, molecular glass formers, ionic liquids) and physiological fluids [4,5,6,7,8,9,10]. Low frequency shear elasticity has profound implications for microfluidics, thermodynamics, and more generally for the understanding of the fluidic transport of the instabilities, thus with possible interest for the industrial and biological sectors. The immediate consequence is that a thermo-elastic coupling is possible, challenging the assumption of an instant dissipation via the fast lifetime of the thermal fluctuations [26].

The existence of shear elasticity in fluids at low frequency justifies the search of another solid property: the thermoelasticity. In solid (isotropic) materials, shear elasticity *G* and bulk elasticity *E* are linked by the Poisson equation, and both *G* and *E* are thermodynamic quantities whose values depend on how the temperature varies during the shear or pressure variation.

We here explored the thermal properties that might be correlated to excitation of (shear) elastic modes. For this, we examined the thermal response of a fluid submitted to an oscillatory shear strain of finite amplitude in accordance with the principle of viscoelastic measurements. A dynamic thermal behavior is highlighted for the three fluids—a polymer melt, a polymer glass former, and a molecular glass former—studied by applying a low frequency oscillatory shear strain.

We show on these three liquid examples, which can be usually considered as ordinary viscous or viscoelastic liquids, that the deformation energy is converted into temperature changes generating non-uniform and complementary thermal shear bands at the sub-millimeter scale. Thus, the internal energy of the liquid changes upon application of mechanical shear stress, demonstrating a necessary coupling with elastic shear modulus. This work also reveals the utmost interest of a thermal approach for the study of liquids.

## 2. Experimental

The infra-red emissivity measurements were carried out in real-time conditions with a microbolometer array of 382 × 288 pixels working at 27 Hz in the range of long wave infrared bands (LWIR) (i.e., wavelengths ranging between 7 to 14 µm). The thermal emissivity is measured by radiation transfer using the Stefan-Boltzmann law: *E = e_m_ σ A* (*T*^4^ − *T*_c_^4^) where *E* is the radiated energy; *e_m_* is the emissivity coefficient; *A* is the radiating area; *T* is the temperature of the sample; and *T_c_* is the temperature of the surroundings. *σ* is the Stefan’s constant. The microbolometer array focuses the liquid surface with a depth field of 0.1 mm and a thermal sensitivity of 0.02 °C, and the liquid is confined between two surfaces: one animated with an oscillatory motion of frequency *ω*, and the other one is fixed. The thermal pictures are corrected from the static thermal environment by subtracting the median value measured at rest prior to the dynamic measurements and collected to lower by a factor ten the noise equivalent temperature difference. The shear strain *γ* is defined by the amplitude of the displacement divided by the gap thickness: *γ = δl/e* where *δl* is the displacement and *e* is the gap thickness. The transmission of the stress from the surface to the sample is reinforced by using high energy alumina fixtures of 45 mm diameter [8,9,27]. The excellent wetting procured by the alumina substrate strengthens the interaction of the liquid molecules to the surface. The high affinity to the substrate reduces the interfacial gas layer trapped between the liquid and the substrate (“pancake” effect) and lowers its propensity to slip on the surface (total wetting conditions). The mechanical stress is provided using a conventional rheometer (ARES2 from TA-Instruments) imposing a sin shape oscillatory shear strain following the conventional formalism: *γ*(*t*) *= γ*_0_*·sin*(*ω·t*) with *γ*_0_ the imposed shear strain coupled to the thermal measurement [8,9]. In the present study, the low frequency regime (*ω* = 1 rad/s) was probed where the fluids exhibited a viscous-like behavior. All the measurements were carried out at room temperature.

The three samples of the study were the following:-The ordinary polymer sample was an amorphous polymer: a polybutyl acrylate (PBuA) of Mn = 40,000 molecular weight and 1.19 polydispersity (Polymer Source Inc. manufacturer, Dorval (Montreal), Quebec). This molecular weight corresponds to the onset of the entanglement state (*Mw* ≅ *2 Me* where *Me* = 22,000) [28]. The melt was studied at room temperature (i.e., at about 100 °C above the glass transition temperature (*Tg* = −64 °C)) with a 0.285 mm gap thickness. Its room temperature terminal viscoelastic time as deduced from conventional viscoelastic measurements was τ_relax_ = 0.03 s [9]. At the low frequency probed for the thermal study (1 rad/s) and macroscopic scale, the PBuA exhibits a viscous response with η = 380 Pa s [6].-The polymer glass former studied was polypropylene glycol-4000 (Sigma-Aldrich manufacturer, St. Louis, MO, USA). The molecular weight is given as 3500–4500 g/mol, which corresponds to about 55 repetition units. It is a viscous liquid at room temperature (*η* = 100 mPa s) and its relaxation time is out of the dynamic range of conventional mechanical tools [29].-Glycerol is a very well-studied liquid due its extremely wide range of uses and its biocompatibility. It is a glass former that exhibits at room temperature of viscosity (*η* = 1.41 Pa s). The molecular relaxation time is far away from the dynamic range of mechanical tools, being accessible by Brillouin scattering at 7 GHz [30].

We restricted the study to the low frequency response (Hz) and at the sub-millimeter scale by using total fluid/substrate wetting conditions to amplify the energy transfer from the surface to the fluid.

## 3. Results

We probed and compared the thermal response of three fluids upon applying a low frequency (1 rad/s) oscillatory shear strain in small gap conditions (*e*~300 µm). We first focused on the low molecular weight polymer melt (PBuA, *Mw* = 40,000), which is characterized by both H-bond (intermolecular) interactions and van der Waals (inter-sidechain) interactions [31]. Then, we present the thermal responses exhibited by the polymer and molecular H-bond glass formers of PPG and glycerol, respectively.

### 3.1. Identification of a Thermal Response of a Polymer Melt (PBuA) to an Oscillatory Shear Deformation

Figure 2 gathers the 2D reconstructed image of the thermal evolution obtained when increasing the shear strain amplitude. At rest, prior to the strain application (Figure 2a), the temperature of the melt is at equilibrium. Upon increasing the shear strain, the polymer melt emits a thermal signal particularly pronounced at high strain rate (Figure 2b–d). The thermal signal splits in cold and hot shear bands that oscillate with the applied strain excitation. At low strain amplitude (*γ* < 200%), the medium band of the gap exhibits a cooling while the fluid band near the surface shows a warming effect. Figure 2e displays the temperature variation in three zones arbitrarily defined as the zones where the temperature variations were the highest. Figure 2e indicates that both cold and hot thermal waves oscillated oppositely and approximately in phase with the strain. Upon increasing the shear strain, cold and hot thermal shear bands reinforced nearly linearly up to *γ* < 100% (Figure 2f). At high shear strain amplitude (*γ* > 500%), the thermal contrast was in some places larger than 1 °C. The thermal shear bands were very visible and complex, superposing multiple hot and cold thermal waves elongated along the strain direction (Figure 2d). 

### 3.2. Identification of a Thermal Response in H-Bond Glass Formers upon Low Frequency Shear Deformation

Viscosity ratios of about 10^3^ and of 10^5^, respectively, separate the polymer melt (PBuA) from polypropylene glycol (PPG-4000) and glycerol (at room temperature). At low frequency (*ω* ~ 1 rad/s) and strain rates ranging between *γ* = 20–2000%, the rheological signature is typically viscous, as shown in Figure 3a and Figure 4a for PPG and glycerol, respectively measured at *ω* = 1 rad/s and *γ* = 1500%.

The following results show that in the same strain-frequency conditions, a thermal effect can be observed, but at higher shear strain amplitudes compared to the polymer melt. At similar thickness and below 200%, no thermal effect was detectable on PPG-4000 and glycerol. The thermal signal emerged progressively at *γ* > 200% and exhibited a nearly linear dependence with the strain amplitude (Figure 2d). For conditions equivalent to the highest strain applied for the polymer melt (Figure 3c measured at 1500% and 1 rad/s), the thermal response of the PPG-4000 displayed a simpler thermal scheme: The liquid splits in alternative cold and hot waves oscillating nearly in phase with the excitation ∆*T*(*t*) = ∆*T_A_·sin*(*ω·t* + ∆*ϕ*) where ∆*T_A_* is the amplitude of the thermal wave and ∆*ϕ* is the phase shift with respect to the shear strain wave, (Δ*ϕ* is about 37 ± 2° in Figure 3c). In contrast to the polymer melt, the thermal signal maintained the shape of the excitation and was stable over a wide strain range (at least up to 1500% at 340 µm).

Figure 3d shows the remarkable linear dependence of the thermal variation with respect to the strain amplitude. This means that cold and hot variations increase symmetrically with the strain rate obeying the sin model oscillating around the equilibrium temperature defined by a zero-∆*T* (here the absolute value of the thermal variation is represented). The quantity reported to the strain value ∆*T_A_*(*γ*)*/γ* is thus constant. It represents the thermal analogue of the shear stress σ reported to the shear strain, *σ/γ*, which defines the shear elastic modulus following Hooke’s law. Therefore, while the viscoelastic measurement indicates a viscous behavior (Figure 3a), the simultaneous recording of the thermal behavior indicates cold and hot thermal waves that evolve synchronously with the strain wave, thus is a non-dissipative behavior. The symmetric evolution of the cold and warm branches of the thermal wave indicates a strict compensation in terms of energy gain and loss that fits with a simple sin model ∆*T*(*t*) = ∆*T_A_·sin*(*ω·t* + ∆*ϕ*) where ∆*ϕ* is always found smaller than π/4. These thermal features point out a non-dissipative behavior indicating that the viscous behavior measured by stress measurements is apparent only and does not highlight the complexity of the liquid dynamic.

Similar thermal features were identified with the glycerol under oscillatory shear strain (at 350 µm). Figure 4b indicates a noisier, weaker signal and exhibiting a different (face to face) hot and cold wave profile for glycerol compared to the alternation of hot and cold zones for PPG-4000. Glycerol is a molecular liquid with symmetrical geometry whereas the low molecular weight PPG-4000 is an oligomer, meaning increased conformational entropy and multiple intermolecular interactions per molecule that might stabilize the fluid to the mechanical deformation. This is also in agreement with a lower “static” shear elasticity in glycerol (i.e., a weaker liquid cohesion) [9].

## 4. Discussion: Remarkable Net Temperature Invariance

The thermal study of these three different liquids away from any phase transition has revealed the importance of a thermal approach. It demonstrates that a low frequency mechanical action modifies the liquid thermal equilibrium. The dynamic thermal changes prove that (ordinary) fluids are endowed with thermoelastic properties, challenging the concept according to which the energy is simply dissipated in fluctuations. The thermal response requires long-range collective interactions, not taken into account in conventional models. Indeed, for excitations of frequency lower than the molecular frequency (typically *ω* < 10^9^–10^12^ Hz for simple liquids), liquids are expected to exhibit a hydrodynamic flow [13,14,17]. Instead, the thermal effects reveal the ability of the liquid to convert the (shear) wave in local thermodynamic states, challenging the assumption of an instant dissipation of a (low frequency) mechanical action in the noise of the thermal fluctuations [32].

This interpretation seems validated by these remarkable features:

The strain-induced temperature oscillated around the equilibrium temperature, which corresponded exactly to the zero displacement (i.e., to the equilibrium temperature) (Figure 5). The time-integration over a period of the temperature showed that the net temperature of the fluid did not change. For such an adiabatic progress to exist, the thermally different waves should not be independent but related. The temperature was globally maintained constant over a period and was equal to the equilibrium temperature (within the error bars); the adiabatic process keeps the bulk volume temperature invariant (and thus the energy).

The thermal waves approximately reproduced the sin shape of the excitation and exhibited a linear dependence to the mechanical shear strain (at low frequency and relatively low strain amplitude), evidencing a direct thermo-strain coupling. Figure 6 details the rapid emergence and the fast relaxation of the thermal signal when the shear strain wave is applied and the motion is stopped.

These properties describe a straightforward conversion of the mechanical strain in thermal states (in the low strain regime). In the same way, the reversibility and the thermal balance within the bulk rule out time-dependent processes such as conduction, diffusivity, or convection effects. A nearly instant and reversible response is a characteristic of the excitation of (shear) elastic property, indicating that thermal and elastic waves are coupled. This elastic energy measures the cohesion energy (i.e., the energy involved in the intermolecular forces). In other words, the thermo-elastic coupling shows that a low frequency mechanical action modifies the intermolecular distances. Thus, hot and cold zones represent slightly shear compressed and dilated (dynamic) states of the liquid that oscillate around the equilibrium zero-strain position that corresponds to the volume at rest.

## 5. Conclusions

A prerequisite for the understanding of the fluidic properties is a complete characterization of the properties prior to entering the flow regime. This pioneering experimental thermal approach highlights a novel property: the ability of mesoscopic fluids to convert the shear energy in a thermal dynamic signal, defining a new type of thermoelasticity. A low molecular weight polybutylcrylate, a polypropylene glycol, and glycerol exhibited strain induced thermal waves synchronous with the applied frequency. These observations were carried out at the mesoscopic scale, away from any phase transition that might indicate a probable generic thermal property of liquids. In this range and according to Deborah number (τ. *ω* < 1), no coupling with viscoelastic (or molecular) relaxation time is expected. The fluid dynamics is thus examined in a regime where the mechanical response is a viscous behavior (flow regime).

The strain-induced cold and hot zones can reach locally thermal amplitudes of ±0.5 °C in the case of the unentangled polymer melt, and about ±0.05 °C for low molecular weight liquids. The generation of cold zones and the low molecular weight of the tested samples exclude an interpretation in terms of viscous friction heating as it is conventionally expected in highly viscous liquids such as entangled polymer melts at high (steady state) shear rates [32,33]. The emergence of cold and hot zones synchronous with the external oscillatory field indicates that no heat exchange enters or leaves the system (in the relatively low strain regime). This adiabatic process highlights that the liquid is able to transform the injected shear strain energy in non-equilibrium temperatures, and thus in dynamic coexisting stretched and compressed states. The thermal waves are thus a visualization of positive and negative stresses induced by the dynamic mechanical shear field.

For such a dynamic thermo-mechanical coupling to exist, it is necessary that liquids support shear stress and thus assume a coupling of shear elastic modes with the bulk elasticity of the liquid, similarly as for solid materials. The thermal waves reveal a collective liquid response (i.e., long range intermolecular interactions). This is in agreement with recent experimental and theoretical results indicating that liquids support shear waves at low scale [11,12,13,14,17,19,20,21,22], and experimentally identified in various simple and complex fluids [3,4,5,6,7,8,9,10]. Low frequency shear elasticity measures the strength of the intermolecular interactions. Liquid shear elasticity makes possible the identification of new non-equilibrium properties such as the strain-driven thermoelastic effects that we are just beginning to discover experimentally [34] and echoes new theoretical predictions for liquid dynamics [11,12,13,14,19,20,21] and liquid thermal balance [35,36,37].

## Figures and Tables

**Figure 1 polymers-13-02378-f001:**
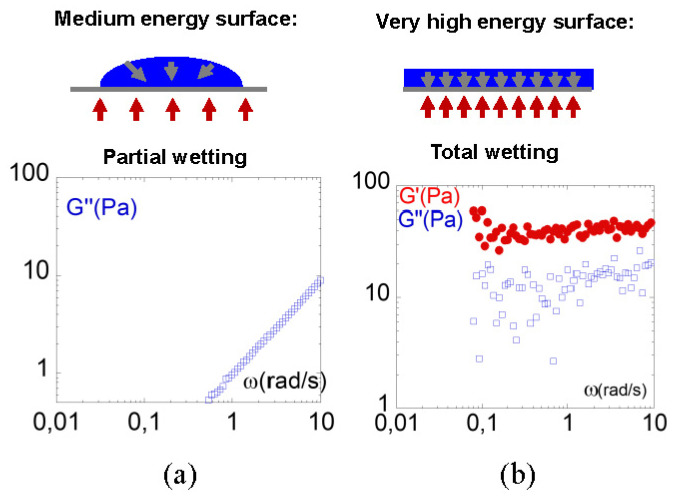
Influence of the wetting on the viscoelastic mesoscopic response of glycerol (from [9]). (**a**) Viscous response obtained on a conventional metallic support (aluminum)—the viscous modulus scales conventionally as *G*” *= η·ω* with *η* = 1 Pa s. (**b**) Higher viscoelastic moduli are obtained using a wetting substrate (alumina) highlighting a solid-like behavior (G’ > G”) at mesoscopic scale (*e* = 40 µm). Room temperature measurements. The upper schemes represent the drop profile in the case of a partial wetting (**a**) and of a total wetting reaching zero contact angle (**b**).

**Figure 2 polymers-13-02378-f002:**
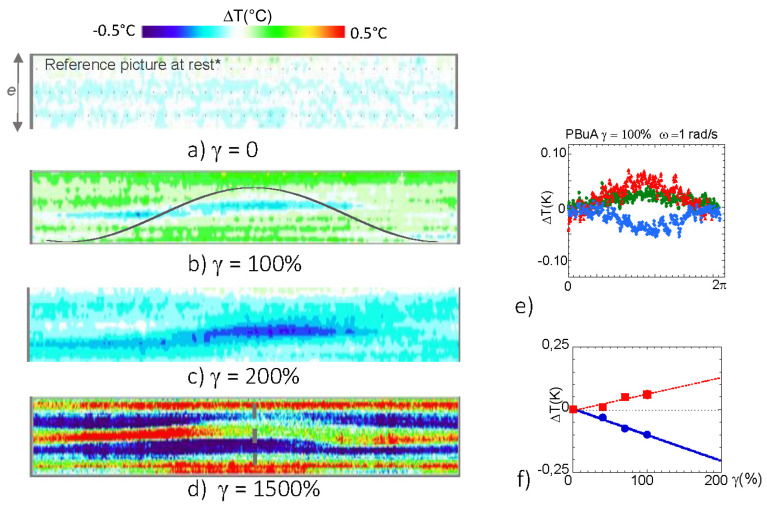
Time evolution of temperature cartography of the low molecular weight polymer melt (PBuA–Mn = 40,000) at rest (**a**) and along one oscillatory period (*ω* = 1 rad/s)(**b**–**d**) submitted to an oscillatory shear strain (280 µm gap, alumina surfaces, room temperature measurements). The upper plane was fixed while the bottom one oscillated. The continuous line of (**b**) is an eye-guide of the sin strain function. (**a**) *γ* = 0%, (**b**) *γ* = 100%, (**c**) *γ* = 200%, (**d**) *γ* = 1500%. (**e**) Temperature profiles over one oscillatory period in the central part (blue points) and on each side of the gap (bottom (red) and upper (dark green) bands) at 100% shear strain and 1 rad/s (the profiles are determined in horizontal bands centered on the highest, the lowest and the medium temperature zones). (**f**) Evolution of the temperature variation (∆*T* = *T* − *T_γ*=0*_*) of the cold (●) and of the hot zones (■). The large strain value (*γ* = 1500%) is not represented due to the occurrence of multiple thermal bands.

**Figure 3 polymers-13-02378-f003:**
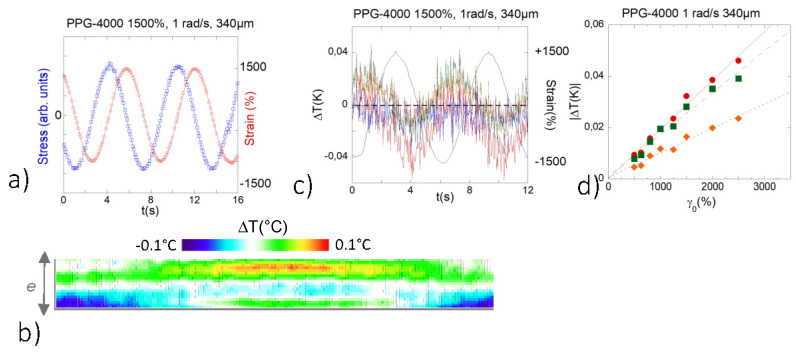
PPG-4000: (**a**) Stress–strain curves for the PPG (**a**) at 340 µm, *ω* = 1 rad/s, and 1500% (measured simultaneously to the thermal measurements; the upper plane was fixed while the bottom one oscillated). (**b**) Thermal mapping of the gap filled with PPG-4000 recorded along one oscillatory period (*ω* = 1 rad/s, *γ* = 1500%, and *e* = 340 µm gap thickness, room temperature measurements carried out on alumina plates, the upper plane was fixed while the bottom one oscillated). (**c**) Details of the temperature oscillation: cold band (■), the hot band (●) and bulk value (♦), black line: strain wave (*ω* = 1 rad/s, *γ* = 1500%, and *e* = 340 µm gap thickness)). (**d**) Strain dependence of the temperature variation (absolute value |∆*T*| = *T* − *T_γ_*_=0_) at 1 rad/s and *e* = 340 µm: cold band (■), the hot band (●), and bulk value (♦) highlighting the linear dependence of the temperature variation to the strain amplitude. The bands are centered on the highest, the lowest and the medium temperature zones.

**Figure 4 polymers-13-02378-f004:**
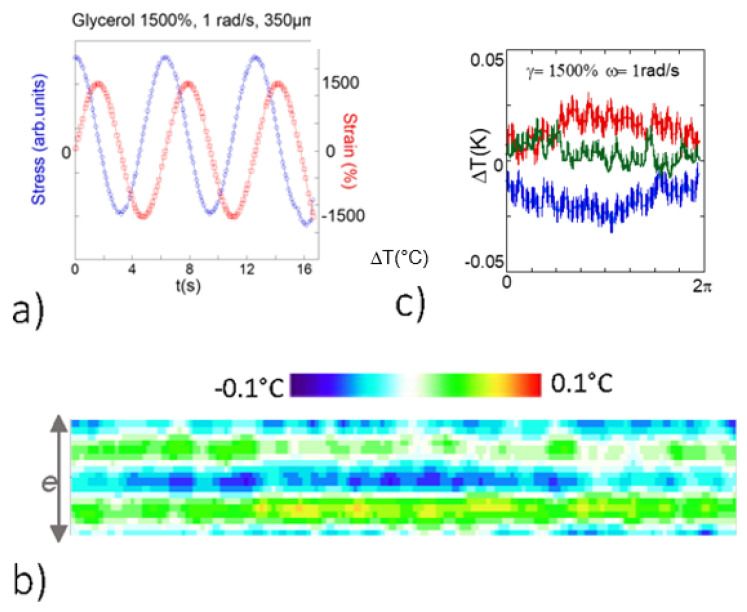
Glycerol: (**a**) Stress–strain curves of glycerol at 350 µm, *ω* = 1 rad/s, and 1500% (simultaneous measurement with the thermal measurements; the upper plane was fixed while the bottom one oscillated). (**b**) Thermal mapping of the glycerol along one oscillatory period (at *γ* = 1500% and *e* = 350 µm gap thickness, alumina surfaces, room temperature measurements, the upper plane was fixed while the bottom one oscillated at *ω* = 1 rad/s). (**c**) Temperature profiles over one period of the temperature recorded on the central part (blue points) and on each side of the gap (bottom red) and upper (dark green) bands) at 1500% shear strain and 1 rad/s.

**Figure 5 polymers-13-02378-f005:**
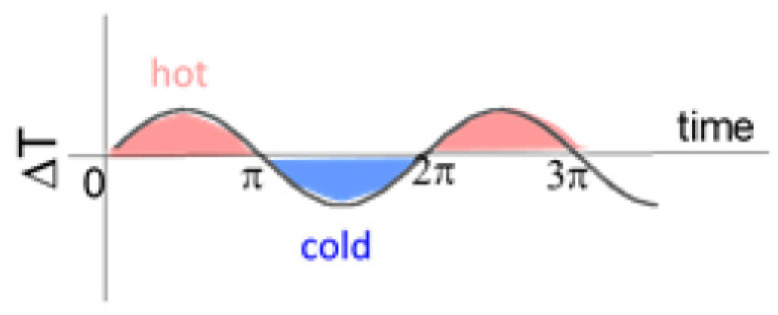
Simplified scheme illustrating the time-dependent thermal wave (single thermal wave) that alternates cold and hot thermodynamic states. The thermal wave, which can be modeled by a sin wave, oscillates around the equilibrium temperature–thermal balance) and is synchronous with the excitation.

**Figure 6 polymers-13-02378-f006:**
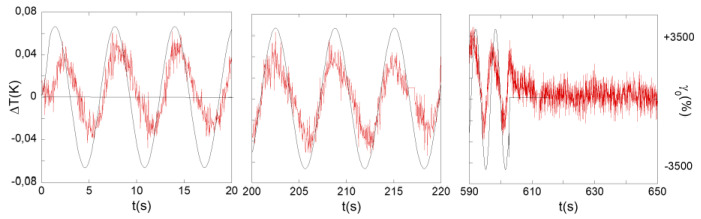
Evolution of the thermal signal of the glycerol (band near the moving surface that presents the highest thermal variation) versus time (*γ* = 1500%, *e* = 350 µm gap thickness, *ω* = 1 rad/s). From left to right: onset of the applied shear strain (measured on 20 s), establishment of the thermal wave, and its relaxation at the stop of the shear strain represented by a continuous black line (measured on 60 s).

## Data Availability

Data are available on request to the authors.

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
