# Peer review of "Highlighting Thermo-Elastic Effects in Confined Fluids"

_polymers, 2021, doi:10.3390/polym13142378_

Round 1

Reviewer 1 Report

This work is interesting, showing that low frequency shear elasticity measures the strength of the intermolecular interactions of polymer melts. At very low shear oscillations close to an adiabatic process, the liquid is able to transform the injected shear strain energy in non-uniform temperatures.

  1. These polymer melts, if looking at the molecular structures, they are all polar molecules. I am not sure how they are called van der Waals fluids, such as PBuA (page 3).
  2. Because there is no phase change in 100mu – mm confinement (is this called low scale?), what about the role of wetting/nonwetting confinement on the nearby molecular structures of the fluids?
  3. I would like to see authors to have a further discussion of simple nonpolar fluids (such as cyclohexane -- PNAS (2018)115 6560-6565) to see at thick film if such thermos-mechanical coupling could be possible.
  4. Why put Fig. 2 before Fig. 1? Section numbers are mixed up: 2.2 – 3.2? More grammatical errors need to be fixed.

Reviewer 2 Report

The manuscript presents convincing experimental evidence for the presence of low-frequency elastic properties in confined liquids with total wetting using thermo-elastic measurements on three different samples. Both the quality of the results and the presentation are good. I suggest to publish the manuscript after taking care of some minor revisions.

  1. The authors state that the presence of elastic properties implies a finite compressibility coefficient. I do not agree. The compressibility coefficient, simply the inverse of the bulk modulus, is finite both in liquids and in solids. Standard bulk liquids have a finite compressibility coefficient which actually determines the speed of longitudinal sound modes therein. As such, the presence of a finite bulk modulus cannot be use in any way to prove any solid-like behaviour in liquids. The big difference between solids and liquids is in the transverse sector, the shear waves that the authors are after. It is important to make this point clear.
  2. In page 7, the sentence before Ref.[20] is not very clear. I suggest the authors to reword it.
  3. Fig.4, contrarily to all other figures, is not centered
  4. Fig.5 also goes out of the margin.
  5. Within the theoretical references cited, I suggest to include Phys. Rev. D 102, 025012 (2020) - Field theory of dissipative systems with gapped momentum states (aps.org) which is a much more detailed extension of Refs.[14-15] and Phys. Rev. Materials 5, 035602 (2021) - Universal ${L}^{\ensuremath{-}3}$ finite-size effects in the viscoelasticity of amorphous systems (aps.org) which completes Ref.[10] with more discussion. The interested reader could benefit of this.
